# Integral Non-Singular Terminal Sliding Mode Consensus Control for Multi-Agent Systems with Disturbance and Actuator Faults Based on Finite-Time Observer

**DOI:** 10.3390/e24081068

**Published:** 2022-08-02

**Authors:** Pu Yang, Yu Ding, Ziwei Shen, Kejia Feng

**Affiliations:** Department of Automation, Nanjing University of Aeronautics and Astronautics, Nanjing 211106, China; jasondy@nuaa.edu.cn (Y.D.); shenziwei@nuaa.edu.cn (Z.S.); fdaxia@nuaa.edu.cn (K.F.)

**Keywords:** integral non-singular terminal sliding mode control (INTSMC), multi-agent systems (MASs), finite-time observer, fault-tolerant control

## Abstract

This paper studies the consensus fault-tolerant control problem of a class of second-order leader–follower multi-agent systems with unknown disturbance and actuator faults, and proposes an integral non-singular terminal sliding mode control algorithm based on a finite-time observer. First, a finite-time disturbance observer was designed based on a combination of high-order sliding mode and dual layers adaptive rules to realize fast estimation and compensation of disturbance and faults. Then, a sliding surface with additional integral links was designed based on the conventional sliding surface, and an integral non-singular terminal sliding mode controller is proposed to realize the robust consensus in finite time and accurately diminish the chattering phenomena. Finally, a numerical example and simulation verify the effectiveness.

## 1. Introduction

With the rapid development of society, communication, artificial intelligence, and so on, multi-agent systems (MASs) have become a hot research issue in the field of control engineering. Due to the collaboration and cooperation of individuals between MASs, these systems can complete complex tasks that are difficult for a single individual. At present, MASs are widely used in mobile sensor networks [1], mobile robots [2], unmanned aerial vehicle formation [3], satellite cluster attitudes [4], and other engineering fields [5,6,7].

In recent years, the problem of consensus control has always been an important research direction in the field of MASs. The goal of consensus control is to ensure that each agent subsystem can use its own local information to complete the task and make the states of all subsystems eventually converge to the same value. Much research has been conducted on consensus control, and this research has produced effective results for different objects in different fields. In [8], an adaptive control protocol is proposed to adjust the virtual impedance to compensate for the mismatched line impedance to achieve accurate reactive power distribution. In [9], a control protocol of general linear MASs is designed by using scroll optimization control. To reduce unnecessary calculation pressure, a reduced-order controller is used to achieve the consensus control of output feedback MASs [10]. In [11], the time delay problem of a class of high-order MASs is considered, and a distributed state feedback control strategy is designed to ensure the consensus of the system.

Some adaptive robust fault-tolerant control methods are proposed to solve the consensus problem of nonlinear leader–follower MASs [12,13,14]. In [15], a static event-triggered protocol is designed to solve the leader–follower consensus problem of first-order MASs with disturbance. In [16], a finite-time fault-tolerant super-twisting algorithm is proposed to solve the effects of actuator faults and unknown disturbance, avoiding the chattering problem. In [17], a novel dynamic sliding mode control protocol is proposed to achieve the finite-time consensus of nonlinear heterogeneous multi-agent systems, which ensures their robustness. In [18], the Gaussian basis function is introduced to deal with the non-strict feedback term, which realizes the leader–follower consensus of multi-agent systems under the unknown switching mechanism.

In fact, MASs are susceptible to disturbance and faults from dynamic models and exosystems during operation. Taking the wheeled robot system as an example, in the actual operation process, the robot will inevitably be affected by external disturbances due to the different road conditions and the surrounding environment and by actuator faults due to incorrect installation or wheel wear. The unknown disturbance and faults will reduce the control effect of the system, bring significant uncertainty, and eventually lead to a decline in the overall performance of the MASs and even the failure of the task. Given that sliding mode control can overcome the uncertainty of the system as well as its strong robustness to interference and unmodeled dynamics, it is widely used in the consensus control of MASs.

Zhao D et al. proposed an adaptive sliding mode control method for a class of second-order systems with a leader to achieve the tracking consensus of the system [19]. Jiang Y L et al. used the integral sliding mode method to achieve a consensus for a class of multi-agent systems with state delays [20]. Dong et al. used the linear sliding mode method to solve the problem of time-varying topology in a second-order nonlinear MASs tracking system [21]. Zhao L et al. studied a class of second-order nonlinear MASs with external disturbance based on the terminal sliding mode method to enable the control system to achieve tracking consistency in a limited time [22]. Sanjoy M et al. studied the consensus of a class of high-order nonlinear MASs based on the integral sliding mode method [23].

Zheng et al. proposed an event-triggered sliding mode method to solve the consensus of a class of leader–follower MASs [24]. Zhao et al. studied the robust consensus of high-order MASs based on distributed protocols, whose considered system models are general, and the designed protocols have strong robustness [25]. However, from the presented results, the chattering phenomenon of this algorithm is not sufficiently weakened. In [26], a distributed fixed-time control algorithm is designed based on backstepping for a class of high-order multi-agent systems, successfully applied to the control of a class of wheeled robot systems. However, the backstepping method needs to continuously differentiate some nonlinear functions, which leads to a high degree of complexity and nonlinearity of the obtained control law, especially when the system order is high.

To solve these problems, the improvements outlined in this paper mainly focus on two aspects. On the one hand, the terminal sliding mode control is selected instead of the backstepping control in the controller design method to overcome the expansion problem of the differential term of the controller. On the other hand, it is necessary to select an appropriate switching function to effectively solve the chattering problem. In addition, the faults and disturbance of the second-order MASs are unpredictable during the operation process, so an observer is the most popular choice to estimate and make up for it based on active control. Zaidi et al. designed a distributed observer to estimate, which takes the external disturbance into account to optimize [27]. To avoid dependence on speed information, Hua et al. proposed two distributed finite-time algorithms that only include a relative position information measurement [28].

Although the idea of an observer is relatively developed, the results of applying it to multi-agent fault diagnosis are relatively scarce, and most of the research stops at fault estimation. In [29], a new controller based on an integral sliding mode and super-twisting sliding mode is designed to solve the fault-tolerant tracking problem of a fault multi-agent system. In [30], based on the idea of the global sliding mode, the disturbance term containing a fault was extracted from the system model, and a distributed observer that can accurately estimate the fault information was designed. Most of the research considers the information of global variables in the design of the observer, and the observer requirements are relatively high, especially for large and complex systems, which should be improved.

The shortcomings of previous research can be summarized in two aspects. On the one hand, some methods combine sliding mode control with other control methods such as adaptive control, which creates a complicated controller with many parameters to be optimized. On the other hand, some methods are sensitive to disturbances and uncertainties, adding unnecessary restrictions or too many parameters with the design. In order to reduce the complexity of the control algorithm and the need for uncertain parameters, this paper, inspired by the existing research, studies the finite-time consensus problem for second-order leader–follower MASs with unknown disturbance and actuator faults by using the integral non-singular terminal sliding mode control algorithm based on the finite-time observer. The main contributions are as follows:A novel adaptive finite-time observer is designed based on a combination of high-order sliding mode and dual-layer adaptive rules, which realizes the centralized estimation and compensation of unknown disturbance and actuator fault in finite time. Additionally, there is no need to obtain the upper bound of the disturbance in advance;On the basis of the conventional sliding mode surface, a new integral part is added to the sliding mode surface, which improves the robustness of the system and sufficiently diminishes the chattering phenomena. The controller proposed in this paper solves the singularity problem and realizes consensus in finite time for disturbed second-order leader–follower MASs;The model studied in this paper comprehensively considers the influence of nonlinear terms, unknown external disturbance, and actuator faults, which improves the practicability of the control algorithm.

The remaining part of this paper is organized as follows: In Section 2, the graph theory and problem formulation are given, and some lemmas, assumptions, definitions, and notations that will be used later are listed. In Section 3, a finite-time observer based on a high-order sliding mode and dual-layer adaptive rule is proposed, and an integral non-singular terminal sliding mode controller is designed, which is analyzed by Lyapunov stability theory. In Section 4, a numerical example and simulation verify the effectiveness of the proposed method in this paper compared with the existing method. Finally, a brief conclusion is given in Section 5.

## 2. Preliminaries and Problem Formulation

### 2.1. Graph Theory

Consider a MAS containing n agents whose topology structure of communication is denoted by the notation G=(V,E,A), where $V=\left\{v_{1},v_{2},\cdots ,v_{n}\right\}$ denotes the set of nodes, E⊆V×V denotes the set of directed edges, and $A=\left[a_{ij}\right]$ is defined as the adjacency matrix. An ordered pair of nodes $\left(v_{j},v_{i}\right)$ is used to represent that a directed edge exists from $v_{j}$ to $v_{i}$, and $v_{i}$ can obtain information from $v_{j}$. If $\left(v_{j},v_{i}\right)\in E$, then $a_{ij}>0$, otherwise $a_{ij}=0$. The main diagonal elements of the adjacency matrix are equal to 0, that is $a_{ii}=0$. The degree of node i is defined as $d_{i}=\sum_{j=1}^{n}a_{ij}$, and $D=diag\left\{d_{1},d_{2},\cdots ,d_{n}\right\}$ denotes the degree matrix. The Laplacian matrix of the graph G is defined as $L=D-A=\left[l_{ij}\right]$, where $l_{ij}$ can be depicted as $l_{ij}=-a_{ij}(j\neq i)$ and $l_{ii}=d_{i}$. For a leader–follower multi-agent system with one leader and n followers, the leader is denoted by node 0, and followers are denoted by node 1,2,⋯,n. The topology structure of communication is denoted by $\bar{G}=\left(\bar{V},\bar{E},\bar{A}\right)$, where $\bar{V}=V\cup \left\{v_{0}\right\}$, $\bar{E}\subseteq \bar{V}\times \bar{V}$, and $\bar{A}$ is the adjacency matrix of $\bar{G}$. Define a matrix $B=diag\left\{b_{1},b_{2},\cdots ,b_{n}\right\}$ to describe whether the leader can directly send information to the follower i, and if it can, then $b_{i}>0$, otherwise $b_{i}=0$. In this paper, the leader can only send information to certain followers, but cannot receive information from any follower.

### 2.2. Problem Formulation

Consider a second-order leader–follower multi-agent system with one leader and n followers. The dynamics of the leader subsystem are described as
(1)$$\{\begin{array}{c} {\dot{x}}_{0}(t)=v_{0}(t)\\ {\dot{v}}_{0}(t)=u_{0}(t) \end{array}$$
where $x_{0}(t)\in R$, $v_{0}(t)\in R$ represent position and velocity state of the leader agent, respectively. $u_{0}(t)\in R$ represents the control input.

The dynamics of the follower i (i=1,2,⋯,n) is described as
(2)$$\{\begin{array}{l} {\dot{x}}_{i}(t)=v_{i}(t)\\ {\dot{v}}_{i}(t)=f_{i}(x_{i}(t),v_{i}(t),t)+u_{i}^{a}(t)+d_{i}(t) \end{array}$$
where $x_{i}(t)\in R$ and $v_{i}(t)\in R$ represent the position and velocity state of the follower agent i. $f_{i}(x_{i}(t),v_{i}(t),t)\in R$ is the inherent nonlinear dynamic function. $d_{i}(t)\in R$ is the external disturbance. $u_{i}^{a}(t)\in R$ denotes the actual control input of the follower agent i, in which the specific model can be expressed as
(3)$$u_{i}^{a}(t)=(1-c_{i})u_{i}(t)$$
where $u_{i}(t)\in R$ denotes the ideal control input, $0\leq c_{i}<1$ denotes the failure factor of the actuator of the follower agent i, and $c_{i}=0$ represents that the control input of the actuator is normal, namely $u_{i}^{a}(t)=u_{i}(t)$. Thus, the system (2) can be rewritten as
(4)$$\{\begin{array}{l} {\dot{x}}_{i}(t)=v_{i}(t)\\ {\dot{v}}_{i}(t)=f_{i}(x_{i}(t),v_{i}(t),t)+u_{i}(t)+\omega_{i}(t) \end{array}$$
where $\omega_{i}(t)=d_{i}(t)-c_{i}u_{i}(t)$ denotes the so-called lumped faults, which include external disturbance and actuator faults.

**Remark** **1.**
*Since the major purpose of the algorithm designed in this paper is to achieve the consensus of MASs, inspired by [31], the actuator failure faults and external disturbances existing in the follower agents are unified into the lumped faults, which are estimated as a whole by the observer. Therefore, during the design process of the control algorithm, there is no need to know the upper limit of the disturbance and the magnitude of the fault, which improves the practicability of the fault-tolerant strategy.*


**Assumption** **1****([32]).***There is a directed spanning tree with the leader as the root node in the graph*$\bar{G}$.

**Assumption** **2**
**([33]).**
*For the nonlinear dynamic function*

$f_{i}(x_{i}(t),v_{i}(t),t)$

*, there is a positive real constant*

$\bar{f}$

*that satisfies*

$\left|f_{i}(x_{i}(t),v_{i}(t),t)\right|\leq {\bar{f}}_{i}$



**Assumption** **3**
**([34]).**
*For the first derivative of the lumped faults*

${\dot{\omega }}_{i}(t)$

*, there is a positive real constant*

$\bar{\omega }$

*which satisfies*

$\left|{\dot{\omega }}_{i}(t)\right|\leq \bar{\omega }$



**Definition** **1**
**([35]).**
*The consensus of the leader–follower MAS is to design a control law*

$u_{i}(t)$

*for each follower, so that the states of the follower tend to those of the leader. The relationship can be described as*

(5)
$$\{\begin{array}{c} \underset{t\to \infty }{\lim}x_{i}(t)-x_{0}(t)=0,\;\forall i=1,2,\cdots ,n\\ \underset{t\to \infty }{\lim}v_{i}(t)-v_{0}(t)=0,\;\forall i=1,2,\cdots ,n \end{array}$$



**Definition** **2****([36]).***The origin points of the system (1) and (2) are considered to be globally finite-time stable if they are globally asymptotically stable with a bounded time function*$T(x_{0})$*, i.e.,*$T_{\max}>0$*such that*$T(x_{0})$*satisfies the term*$T(x_{0})<T_{\max}$.

### 2.3. Some Lemmas and Notations

Some notations are given as follows. The given values $x={\left[x_{1},x_{2},\cdots ,x_{n}\right]}^{T}\in R^{n}$ and γ∈R, denote $\mathrm{sgn}(x)={\left[\mathrm{sgn}(x_{1}),\mathrm{sgn}(x_{2}),\cdots ,\mathrm{sgn}(x_{n})\right]}^{T}$, and $sig^{\gamma }(x)={\left[sig^{\gamma }(x_{1}),sig^{\gamma }(x_{2}),\cdots ,sig^{\gamma }(x_{n})\right]}^{T}$, where sgn(⋅) is the standard signum function and $sig^{\gamma }(x_{i})={\left|x_{i}\right|}^{\gamma }\mathrm{sgn}(x_{i})\;\;i=1,2,\cdots n$. |⋅| is the absolute value. The value $1_{n}$ is the unit column vector in $R^{n}$.

**Lemma** **1**
**([33]).**
*If there is a directed spanning tree in the graph*

$\bar{G}$

*, and the leader is the root node, then the matrix*

L+B

*is invertible.*


**Lemma** **2**
**([37]).**
*Consider a differential equation*

(6)
$$\dot{x}=-\frac{2\alpha_{0}}{1+e^{-\eta_{0}(\left|x\right|-\varepsilon_{0})}}-\frac{2\beta_{0}}{1+e^{\mu_{0}(\left|x\right|-\varepsilon_{0})}}sig^{\gamma_{0}}(x)$$

*where*

$\alpha_{0}$

*,*

$\beta_{0}$

*,*

$\eta_{0}$

*,*

$\mu_{0}$

*are positive constants,*

$0<\gamma_{0}<1$

*, and*

$\varepsilon_{0}={\left(\beta_{0}/\alpha_{0}\right)}^{1/(1-\gamma_{0})}$

*.*

*Accordingly, the dynamic (6) is declared as finite-time stabilization with respect to the initial term*

x(0)

*and the settling time*

$T_{0}$

*is given by*

(7)
$$T_{0}<\frac{\ln(\left|x(0)\right|)-\ln(\varepsilon_{0})}{\alpha_{0}}+\frac{1}{\beta_{0}(1-\gamma_{0})}{\left|\varepsilon_{0}\right|}^{1-\gamma_{0}}$$



**Lemma** **3****([28,38]).***Consider a nonlinear system*$\dot{x}=f(x)$*, where*f(0)=0*. If there is a positive definite continuous function*V(x)*, which makes an open area at the origin*x=0*satisfy*(8)$$\dot{V}(x)+mV^{\delta }(x)\leq 0$$*where*m>0*and*0<δ<1, *then the function*V(x)*will converge to the origin within a certain finite time, and the upper bound of the finite convergence time*T*depends on the initial state of the system*x(0)*, namely*(9)$$T\left(x(0)\right)\leq \frac{V{\left(x(0)\right)}^{\left(1-\delta \right)}}{m(1-\delta )}$$

**Lemma** **4****([39]).***For*$x_{i}\in R(i=1,2,\cdots ,n)$*,*0<η<2*, then*$\sum_{i=1}^{n}{\left|x_{i}\right|}^{{}^{\eta }}\geq {\left(\sum_{i=1}^{n}x_{i}^{2}\right)}^{\frac{\eta }{2}}$.

## 3. Main Results

### 3.1. Design of Consensus

According to the neighbor information obtained by the follower i, the consensus error of position $e_{xi}(t)$ and consensus error of velocity $e_{vi}(t)$ are defined as
(10)$$\{\begin{array}{c} e_{xi}(t)=\sum_{j=1}^{n}a_{ij}\left(x_{i}(t)-x_{j}(t)\right)+b_{i}\left(x_{i}(t)-x_{0}(t)\right),\\ e_{vi}(t)=\sum_{j=1}^{n}a_{ij}\left(v_{i}(t)-v_{j}(t)\right)+b_{i}\left(v_{i}(t)-v_{0}(t)\right), \end{array}\;i\in \left\{1,2,\cdots ,n\right\}$$

Define
$${\bar{x}}_{i}(t)=x_{i}(t)-x_{0}(t)$$
$${\bar{v}}_{i}(t)=v_{i}(t)-v_{0}(t)$$
$$\bar{x}={\left[{\bar{x}}_{1}(t),{\bar{x}}_{2}(t),\cdots ,{\bar{x}}_{n}(t)\right]}^{T}$$
$$\bar{v}={\left[{\bar{v}}_{1}(t),{\bar{v}}_{2}(t),\cdots ,{\bar{v}}_{n}(t)\right]}^{T}$$
$$e_{x}={\left[e_{x1},e_{x2},\cdots ,e_{xn}\right]}^{T}$$
$$e_{v}={\left[e_{v1},e_{v2},\cdots ,e_{vn}\right]}^{T}$$

Then the global synchronization error can be defined as
(11)$$\{\begin{array}{c} e_{x}=(L+B)\bar{x}\\ e_{v}=(L+B)\bar{v} \end{array}$$

Taking the first derivative of $e_{xi}$ and $e_{vi}$ with respect to time, we can obtain
(12)$$\{\begin{array}{ll} {\dot{e}}_{xi} & =e_{vi}\\ {\dot{e}}_{vi} & =\left(\sum_{j=1;j\neq i}^{n}a_{ij}+b_{i}\right)u_{i}-\sum_{j=1;j\neq i}^{n}a_{ij}u_{j}-b_{i}u_{0}\\ & +\left(\sum_{j=1;j\neq i}^{n}a_{ij}+b_{i}\right)\left(f_{i}+\omega_{i}\right)-\sum_{j=1;j\neq i}^{n}a_{ij}(f_{j}+\omega_{j}) \end{array}$$

Define
$$F={\left[f_{1}\left(x_{1}(t),v_{1}(t),t\right),\cdots ,f_{n}\left(x_{n}(t),v_{n}(t),t\right)\right]}^{T}$$
$$u={\left[u_{1}(t),u_{2}(t),\cdots ,u_{n}(t)\right]}^{T}$$
$$\omega ={\left[\omega_{1}(t),\omega_{2}(t),\cdots ,\omega_{n}(t)\right]}^{T}$$
$${\dot{e}}_{x}={\left[{\dot{e}}_{x1},{\dot{e}}_{x2},\cdots ,{\dot{e}}_{xn}\right]}^{T}$$
$${\dot{e}}_{v}={\left[{\dot{e}}_{v1},{\dot{e}}_{v2},\cdots ,{\dot{e}}_{vn}\right]}^{T}$$

Taking the first derivative of Equation (11) with respect to time, the global form of (12) can be obtained as
(13)$$\{\begin{array}{l} {\dot{e}}_{x}=e_{v}\\ {\dot{e}}_{v}=(L+B)(F+u-1_{n}u_{0}+\omega ) \end{array}$$

If the Equation (13) is asymptotically stable according to Definition 1, then the consensus problem of MASs described as (1) and (2) can be solved.

### 3.2. Design of the Finite-Time Observer

Given that the information of velocity and faults is difficult to obtain during the operation of MASs, a novel finite-time observer was designed based on high-order sliding mode observer and dual-layer adaptive rule.
(14)$$\{\begin{array}{c} {\dot{\hat{v}}}_{i}=f_{i}+u_{i}+{\hat{\omega }}_{i}-\frac{2\alpha_{1}}{1+e^{-\eta_{1}(\left|{\tilde{v}}_{i}\right|-\varepsilon_{1})}}{\tilde{v}}_{i}-\frac{2\beta_{1}}{1+e^{\mu_{1}(\left|{\tilde{v}}_{i}\right|-\varepsilon_{1})}}sig^{\gamma_{1}}({\tilde{v}}_{i})\\ {\dot{\hat{\omega }}}_{i}=-\frac{2\alpha_{2}}{1+e^{-\eta_{2}(\left|{\tilde{\omega }}_{i}\right|-\varepsilon_{2})}}{\tilde{\omega }}_{i}-\frac{2\beta_{2}}{1+e^{\mu_{2}({\tilde{\omega }}_{i}-\varepsilon_{2})}}sig^{\gamma_{2}}({\tilde{\omega }}_{i})-\Xi \mathrm{sgn}({\tilde{\omega }}_{i}) \end{array}$$
where ${\hat{v}}_{i}$ and ${\hat{\omega }}_{i}$ are the approximated values of $v_{i}$ and $\omega_{i}$. Denote $\tilde{v}=\hat{v}-v$ and $\tilde{\omega }=\hat{\omega }-\omega$. $\alpha_{k}$, $\beta_{k}$, $\eta_{k}$, $\mu_{k}$ are positive constants, $0<\gamma_{k}<1$, and $\varepsilon_{k}={\left(\beta_{k}/\alpha_{k}\right)}^{1/(1-\gamma_{k})}$, where k=1,2. Additionally, Ξ is an adaptive gain value.

**Theorem** **1.**
*For the MASs described as (1) and (2), if the observer is designed as (14) and the adaptive gain value satisfies*

$\Xi >\left|{\dot{\omega }}_{i}\right|$

*, then the approximation error will converge to zero in finite time.*


**Proof.** Taking the first derivative of ${\tilde{v}}_{i}$ and ${\tilde{\omega }}_{i}$ based on (14), the following can be obtained
(15)$$\{\begin{array}{l} {\dot{\tilde{v}}}_{i}={\tilde{\omega }}_{i}-\frac{2\alpha_{1}}{1+e^{-\eta_{1}(\left|{\tilde{v}}_{i}\right|-\varepsilon_{1})}}{\tilde{v}}_{i}-\frac{2\beta_{1}}{1+e^{\mu_{1}(\left|{\tilde{v}}_{i}\right|-\varepsilon_{1})}}sig^{\gamma_{1}}({\tilde{v}}_{i})\\ {\dot{\tilde{\omega }}}_{i}=-{\dot{\omega }}_{i}-\Xi \mathrm{sgn}({\tilde{\omega }}_{i})-\frac{2\alpha_{2}}{1+e^{-\eta_{2}(\left|{\tilde{\omega }}_{i}\right|-\varepsilon_{2})}}{\tilde{\omega }}_{i}-\frac{2\beta_{2}}{1+e^{\mu_{2}({\tilde{\omega }}_{i}-\varepsilon_{2})}}sig^{\gamma_{2}}({\tilde{\omega }}_{i}) \end{array}$$ A Lyapunov function $V_{1i}=\frac{1}{2}{\tilde{\omega }}^{2}$ can be defined, and the first derivative of $V_{1i}$ with respect to time can be obtained as follows:(16)$$\begin{array}{l} {\dot{V}}_{1i}={\tilde{\omega }}_{i}{\dot{\tilde{\omega }}}_{i}\\ ={\tilde{\omega }}_{i}\left(-{\dot{\omega }}_{i}-\Xi \mathrm{sgn}({\tilde{\omega }}_{i})-\frac{2\alpha_{2}}{1+e^{-\eta_{2}(\left|{\tilde{\omega }}_{i}\right|-\varepsilon_{2})}}{\tilde{\omega }}_{i}-\frac{2\beta_{2}}{1+e^{\mu_{2}({\tilde{\omega }}_{i}-\varepsilon_{2})}}{\left|{\tilde{\omega }}_{i}\right|}^{\gamma_{2}}\mathrm{sgn}({\tilde{\omega }}_{i})\right)\\ =-{\dot{\omega }}_{i}\tilde{\omega }-\Xi \left|{\tilde{\omega }}_{i}\right|-\frac{2\alpha_{2}}{1+e^{-\eta_{2}(\left|{\tilde{\omega }}_{i}\right|-\varepsilon_{2})}}{\tilde{\omega }}_{i}^{2}-\frac{2\beta_{2}}{1+e^{\mu_{2}({\tilde{\omega }}_{i}-\varepsilon_{2})}}{\left|{\tilde{\omega }}_{i}\right|}^{\gamma_{2}+1}\\ \leq -\left(\Xi -\left|{\dot{\omega }}_{i}\right|\right)\left|{\tilde{\omega }}_{i}\right|-\frac{2\alpha_{2}}{1+e^{-\eta_{2}(\left|{\tilde{\omega }}_{i}\right|-\varepsilon_{2})}}{\tilde{\omega }}_{i}^{2}-\frac{2\beta_{2}}{1+e^{\mu_{2}({\tilde{\omega }}_{i}-\varepsilon_{2})}}{\left|{\tilde{\omega }}_{i}\right|}^{\gamma_{2}+1}\\ \leq -\frac{2\alpha_{2}}{1+e^{-\eta_{2}(\left|{\tilde{\omega }}_{i}\right|-\varepsilon_{2})}}{\tilde{\omega }}_{i}^{2}-\frac{2\beta_{2}}{1+e^{\mu_{2}({\tilde{\omega }}_{i}-\varepsilon_{2})}}{\left|{\tilde{\omega }}_{i}\right|}^{\gamma_{2}+1}\\ \leq 0 \end{array}$$According to Lemma 2, the proposed observer can estimate the lumped faults in a finite time. □

The convergence time $T_{1}$ satisfies
(17)$$T_{1}<\frac{\ln(\left|{\tilde{\omega }}_{i}(0)\right|)-\ln(\varepsilon_{2})}{\alpha_{2}}+\frac{1}{\beta_{2}(1-\gamma_{2})}{\left|\varepsilon_{2}\right|}^{1-\gamma_{2}}$$ The design of Ξ depends on the upper bound of $\omega_{i}$ according to Assumption 3, while in the actual application process, it is often accompanied by uncertain parameters affecting the system. To address the need for all uncertain parameters, a dual layers adaptive law is designed to improve the design of Ξ as follows:(18)$$\{\begin{array}{c} \dot{\Xi }=-(\Delta_{1}+\Delta_{2})\mathrm{sgn}(\Psi )\\ {\dot{\Delta }}_{2}=\{\begin{array}{cc} \Delta_{d}\left|\Psi \right|, & \left|\Psi \right|>\Psi_{0}\\ 0, & \left|\Psi \right|\leq \Psi_{0} \end{array} \end{array}$$
where
(19)$$\{\begin{array}{cc} \Psi & =\Xi -\frac{\left|\Lambda \right|}{\kappa_{1}}-\kappa_{2}\\ \dot{\Lambda } & =\zeta fal\left(-\Xi \mathrm{sgn}(\tilde{\omega })-\Lambda ,\chi ,\varsigma \right)\\ fal(\Gamma ,\chi ,\varsigma ) & =\{\begin{array}{cc} {\left|\Gamma \right|}^{\chi }\mathrm{sgn}(\Gamma ), & \left|\Gamma \right|>\varsigma \\ \frac{\Gamma }{\varsigma^{1-\chi }}, & \left|\Gamma \right|\leq \varsigma \end{array} \end{array}$$
where $\Delta_{1}$, $\Delta_{d}$, ζ, $\kappa_{1}$, $\kappa_{2}$, χ and ς are positive constants. Therefore, the value of $\Xi \mathrm{sgn}(\tilde{\omega })$ can be obtained by the fal(⋅) function in real time.

### 3.3. Design of Sliding Mode Controller

From the consensus errors given in Section 3.1, the sliding surface can be described as [40]:(20)$$s_{i}=e_{xi}+\int_{0}^{t}\left(\xi_{1}\mathrm{sgn}(e_{vi}){\left|e_{vi}\right|}^{\theta_{1}}+\xi_{2}\mathrm{sgn}(e_{xi}){\left|e_{xi}\right|}^{\theta_{2}}\right)dt$$
where $\xi_{1}$ and $\xi_{2}$ are positive constants, $1<\theta_{1}<2$, and $\theta_{2}=\theta_{1}/\left(2-\theta_{1}\right)$. To ensure that the convergence time of s→0 is limited, and to eliminate the chattering problem caused by sliding mode control, a novel integral non-singular terminal sliding mode control (INTSMC) surface was designed as follows:(21)$$\begin{array}{ll} \sigma_{i} & ={\dot{s}}_{i}+\lambda s_{i}\\ & =e_{vi}+\xi_{1}\mathrm{sgn}(e_{vi}){\left|e_{vi}\right|}^{\theta_{1}}+\xi_{2}\mathrm{sgn}(e_{xi}){\left|e_{xi}\right|}^{\theta_{2}}\\ & +\lambda \left(e_{xi}+\int_{0}^{t}\left(\xi_{1}\mathrm{sgn}(e_{vi}){\left|e_{vi}\right|}^{\theta_{1}}+\xi_{2}\mathrm{sgn}(e_{xi}){\left|e_{xi}\right|}^{\theta_{2}}\right)dt\right) \end{array}$$
where λ>0 is a tuning constant. The derivative of σ can be obtained as
(22)$$\begin{array}{ll} {\dot{\sigma }}_{i} & ={\ddot{s}}_{i}+\lambda {\dot{s}}_{i}\\ & ={\dot{e}}_{vi}+\xi_{1}\theta_{1}{\dot{e}}_{vi}{\left|e_{vi}\right|}^{\theta_{1}-1}+\xi_{2}\theta_{2}e_{vi}{\left|e_{xi}\right|}^{\theta_{2}-1}\\ & +\lambda \left(e_{vi}+\xi_{1}\mathrm{sgn}(e_{vi}){\left|e_{vi}\right|}^{\theta_{1}}+\xi_{2}\mathrm{sgn}(e_{xi}){\left|e_{xi}\right|}^{\theta_{2}}\right) \end{array}$$

Instituting (12) into (22), we can obtain
(23)$$\begin{array}{ll} \dot{\sigma } & =\left(\begin{array}{l} \left(\sum_{j=1;j\neq i}^{n}a_{ij}+b_{i}\right)u_{i}-\sum_{j=1;j\neq i}^{n}a_{ij}u_{j}-b_{i}u_{0}\\ +\left(\sum_{j=1;j\neq i}^{n}a_{ij}+b_{i}\right)\left(f_{i}+\omega_{i}\right)-\sum_{j=1;j\neq i}^{n}a_{ij}(f_{j}+\omega_{j}) \end{array}\right)(1+\xi_{1}\theta_{1}{\left|e_{vi}\right|}^{\theta_{1}-1})\\ & +\xi_{2}\theta_{2}e_{vi}{\left|e_{xi}\right|}^{\theta_{2}-1}+\lambda \left(e_{vi}+\xi_{1}\mathrm{sgn}(e_{vi}){\left|e_{vi}\right|}^{\theta_{1}}+\xi_{2}\mathrm{sgn}(e_{xi}){\left|e_{xi}\right|}^{\theta_{2}}\right) \end{array}$$

The controller for the follower agent i is designed as follows:(24)$$u_{i}=u_{i1}+u_{i2}$$
with
(25)$$u_{i1}={\left(d_{i}+b_{i}\right)}^{-1}\left\{\begin{array}{l} \sum_{j=1;j\neq i}^{n}a_{ij}u_{j}+b_{i}u_{0}-(2n+b_{i})(\bar{f}+{\hat{\omega }}_{i})+{\left(1+\xi_{1}\theta_{1}{\left|e_{vi}\right|}^{\theta_{1}-1}\right)}^{-1}\\ \xi_{2}\theta_{2}e_{vi}{\left|e_{xi}\right|}^{\theta_{2}-1}+\lambda \left(e_{vi}+\xi_{1}\mathrm{sgn}(e_{vi}){\left|e_{vi}\right|}^{\theta_{1}}+\xi_{2}\mathrm{sgn}(e_{xi}){\left|e_{xi}\right|}^{\theta_{2}}\right) \end{array}\right\}$$
(26)$$u_{i2}=-{\left(d_{i}+b_{i}\right)}^{-1}{\left(1+\xi_{1}\theta_{1}{\left|e_{vi}\right|}^{\theta_{1}-1}\right)}^{-1}\rho \mathrm{sgn}(\sigma )$$
where ρ is a positive constant, $\bar{f}$ is the upper bound of f based on Assumption 2. $u_{i1}$ denotes the part of equivalent control and $u_{i2}$ denotes the part of switching control.

**Theorem** **2.**
*The origin errors of MASs described as (13) can converge to zero in a finite time by the controller designed as (24)−(26), which means the fault-tolerant consensus goals can be achieved.*


**Proof.** A Lyapunov function $V_{2}=\frac{1}{2}\sum_{i=1}^{n}\sigma_{i}^{2}$ is defined, and the first derivative of $V_{2}$ with respect to time can be obtained as follows:
(27)$$\begin{array}{ll} {\dot{V}}_{2} & =\sum_{i=1}^{n}\sigma_{i}{\dot{\sigma }}_{i}\\ & =\sum_{i=1}^{n}\sigma_{i}\left(\left(\sum_{j=1;j\neq i}^{n}a_{ij}+b_{i}\right)u_{i}-\sum_{j=1;j\neq i}^{n}a_{ij}u_{j}-b_{i}u_{0}\right)(1+\xi_{1}\theta_{1}{\left|e_{vi}\right|}^{\theta_{1}-1})\\ & +\sum_{i=1}^{n}\sigma_{i}\left(\left(\sum_{j=1;j\neq i}^{n}a_{ij}+b_{i}\right)\left(f_{i}+\omega_{i}\right)-\sum_{j=1;j\neq i}^{n}a_{ij}(f_{j}+\omega_{j})\right)(1+\xi_{1}\theta_{1}{\left|e_{vi}\right|}^{\theta_{1}-1})\\ & +\sum_{i=1}^{n}\sigma_{i}\left(\xi_{2}\theta_{2}e_{vi}{\left|e_{xi}\right|}^{\theta_{2}-1}+\lambda \left(e_{vi}+\xi_{1}\mathrm{sgn}(e_{vi}){\left|e_{vi}\right|}^{\theta_{1}}+\xi_{2}\mathrm{sgn}(e_{xi}){\left|e_{xi}\right|}^{\theta_{2}}\right)\right) \end{array}$$Introducing (24)–(26) into (27) can transform it into
(28)$$\begin{array}{ll} {\dot{V}}_{2} & =\sum_{i=1}^{n}\sigma_{i}\left(\left(\sum_{j=1;j\neq i}^{n}a_{ij}+b_{i}\right)\left(f_{i}+\omega_{i}\right)-\sum_{j=1;j\neq i}^{n}a_{ij}(f_{j}+\omega_{j})\right)\\ & -\sum_{i=1}^{n}\sigma_{i}((2n+b_{i})(\bar{f}+{\hat{\omega }}_{i}))-\sum_{i=1}^{n}\sigma_{i}\rho \mathrm{sgn}(\sigma )(1+\xi_{1}\theta_{1}{\left|e_{vi}\right|}^{\theta_{1}-1})\\ & \leq \sum_{i=1}^{n}(2n+b_{i})(f_{i}-\bar{f})\left|\sigma_{i}\right|+\sum_{i=1}^{n}(2n+b_{i})(\omega_{i}-{\hat{\omega }}_{i})\left|\sigma_{i}\right|-\sum_{i=1}^{n}\sigma_{i}\rho \mathrm{sgn}(\sigma )\\ & \leq -\sum_{i=1}^{n}\rho \left|\sigma_{i}\right|\\ & \leq -\rho {\left(\sum_{i=1}^{n}\sigma_{i}^{2}\right)}^{\frac{1}{2}}\\ & =-\sqrt{2}\rho V_{2}^{\frac{1}{2}}\\ & <0 \end{array}$$According to Lemma 3, the INTSMC surface designed as (21) can converge to zero in a finite time. □

The convergence time $T_{2}$ satisfies
(29)$$T_{2}\leq \frac{\sqrt{2}}{\rho }{\left(V_{2}(0)\right)}^{\frac{1}{2}}$$

**Remark** **2.**
*Consider the finite convergence time*

$T_{1}$

*and*

$T_{2}$

*, both of which have upper bounds related to the initial states. The value of*

$T_{1}$

*is determined by the initial estimation error of the observer, and the value of*

$T_{2}$

*is determined by the initial consensus tracking error of the follower agents. In practical applications, the initial value of the latter is larger, and thus the convergence time is longer.*


## 4. Simulations

In this section, we will take a numerical simulation to verify the effectiveness and superiority of the method proposed in this paper compared with the method proposed in [30,40]. Consider a leader–follower MAS with one leader and five followers, whose topology structure graph is shown in Figure 1.

The weight of all edges is 1, and the Laplacian matrix L and adjacency matrix B can be calculated according to Figure 1 as
(30)$$L=\left[\begin{array}{lllll} 0 & 0 & 0 & 0 & 0\\ 0 & 0 & 0 & 0 & 0\\ 0 & 0 & 1 & -1 & 0\\ -1 & 0 & 0 & 2 & -1\\ -1 & -1 & 0 & 0 & 2 \end{array}\right],\;B=\left[\begin{array}{lllll} 1 & 0 & 0 & 0 & 0\\ 0 & 1 & 0 & 0 & 0\\ 0 & 0 & 0 & 0 & 0\\ 0 & 0 & 0 & 0 & 0\\ 0 & 0 & 0 & 0 & 0 \end{array}\right]$$

The dynamic function of the leader agent is described as
(31)$$\{\begin{array}{l} {\dot{x}}_{0}(t)={\dot{v}}_{0}(t)\\ {\dot{v}}_{0}(t)=\sin(t/12) \end{array}$$
where the values of initial states are set to $x_{0}(0)=0$ and $v_{0}(0)=1$. The dynamic equation of follower i (i=1,2,⋯,5) is described as:(32)$$\{\begin{array}{l} x_{i}(t)=v_{i}(t)\\ v_{i}(t)=\cos\left(x_{i}(t)\right)+\cos\left(v_{i}(t)\right)+u_{i}^{a}(t)+d_{i}(t) \end{array}$$
where $f_{i}(x_{i}(t),v_{i}(t),t)=\cos\left(x_{i}(t)\right)+\cos\left(v_{i}(t)\right)$ is the inherent nonlinear dynamics. It is apparent that $f_{i}(x_{i}(t),v_{i}(t),t)\leq 2$, which satisfies Assumption 2. The values of the initial states are set to $x_{1}(0)=1$, $x_{2}(0)=1.5$, $x_{3}(0)=-2.2$, $x_{4}(0)=-1.5$, $x_{5}(0)=-0.5$, $v_{1}(0)=1.2$, $v_{2}(0)=0.5$, $v_{3}(0)=0$, $v_{4}(0)=0.8$, $v_{5}(0)=1.5$. Here, we consider that the follower 1 and the follower 4 are with actuator faults and disturbance, while the others are only with disturbance. The failure factors are set to $c_{1}=0.2$ and $c_{4}=0.4$. The disturbance suffered by followers is described as follows
(33)$$d_{1,2}=0.2+0.2\sin(0.5\pi t),\;d_{3,4,5}=0.3+0.1\sin(2\pi t)$$

The tuning parameters are mainly selected based on our design experience and experimental debugging, and a set of parameters that can accurately reflect the tracking and estimation effectiveness of the observer (14) is chosen. The parameter selections of the observers are as followers: $\alpha_{1}=\alpha_{2}=6$, $\beta_{1}=\beta_{2}=6$, $\eta_{1}=\eta_{2}=0.9$, $\mu_{1}=\mu_{2}=1.2$, $\gamma_{1}=\gamma_{2}=0.7$, $\Delta_{1}$ = 5, $\Delta_{d}=20$, $\Psi_{0}=0.7$, ζ=4, $\kappa_{1}=0.4$, $\kappa_{2}=3$, χ=0.9, ς=0.7. The simulations of the lumped faults tracking curve are seen in Figure 2 and Figure 3, and the estimated time of $\omega_{1}$ and $\omega_{4}$ is shown in Table 1. To verify the superiority of the observer proposed in this paper, this result is compared with the observer proposed in [30], which can be seen in Figure 2 and Figure 3.

From Figure 2 and Figure 3, it can be seen that both the observer proposed in this paper and that in [30] have a good estimation. However, it is apparent from Table 1 that the observer proposed in this paper has a better state tracking accuracy and shorter time than that in [30], which verifies that the observer proposed in this paper is able to achieve the consistent tracking of lumped faults in finite time. Therefore, this can help improve performance and reduce dynamic computing load through a timely and accurate estimation.

The choice of parameter settings of the proposed controller is shown below: $\xi_{1}=0.2$, $\xi_{2}=0.2$, $\theta_{1}=1.001$, λ=10 and ρ=25. The simulations of the consensus error curve are seen in Figure 4, Figure 5, Figure 6 and Figure 7. Figure 4 and Figure 5 show the state tracking errors of followers by the controller proposed in this paper. Figure 6 and Figure 7 show the state tracking errors of followers by the protocol in [40]. It can be seen that the tracking error of the position and velocity variables can converge to zero within a finite time by using the controller outlined in this paper. From the comparison of Figure 4 and Figure 5 with Figure 6 and Figure 7, the settling time with the controller in this paper is about 6 s, while that with the protocol in [40] is about 11 s. The overshoot is smaller, and the convergence speed is faster of the curve based on the controller proposed in this paper than that in [40], which proves the superiority of our proposed controller.

The curve of the control input $u_{i}(t)$ by the proposed controller and the protocol in [40] are seen in Figure 8 and Figure 9. The curve of the control input in this paper is smoother and better at eliminating the chattering problem than that in [40].

From the above simulation comparison results, the proposed controller can effectively solve the problem of chattering, while the controller in [40] cannot. Consequently, the control method proposed in this paper has an improved robustness and can realize rapid and accurate control of consensus.

## 5. Conclusions

To solve the consensus problem of disturbed second-order leader–follower MASs with disturbance and actuator faults, this paper proposes a novel integral non-singular terminal sliding mode control algorithm based on a finite-time observer. The addition of an integral link makes the system states converge faster than the traditional terminal SMC and solves the chattering problem. Additionally, the proposed controller performs well under unknown disturbances and actuator faults. Through the verification of a numerical example and simulation, leader–follower MASs can realize consensus under disturbance and actuator faults in finite time, which proves that the algorithm proposed in this paper can effectively improve the robust consensus of the system.

Owing to the superiority of the proposed algorithm and its ability to handle the disturbances and actuator faults, it can be applied in multi-wheeled robotic systems and multi-UAV systems. Their structure is similar to that of the system model described in this study, but minor changes are required in practical applications. In future research, the consensus control of MASs with unmatched disturbance or sensor faults will be considered.

## Figures and Tables

**Figure 1 entropy-24-01068-f001:**
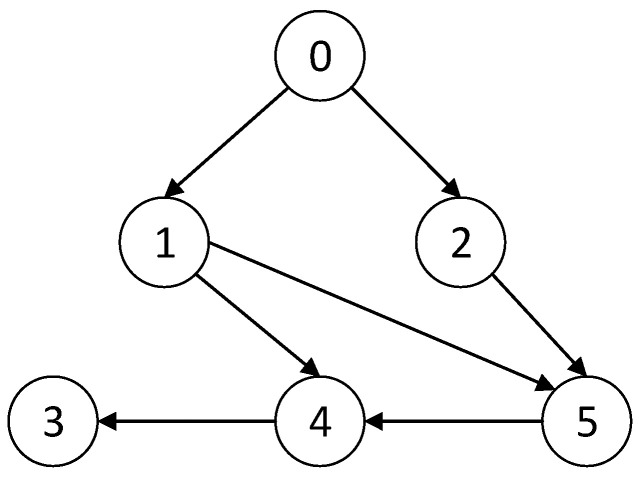
The topology structure graph.

**Figure 2 entropy-24-01068-f002:**
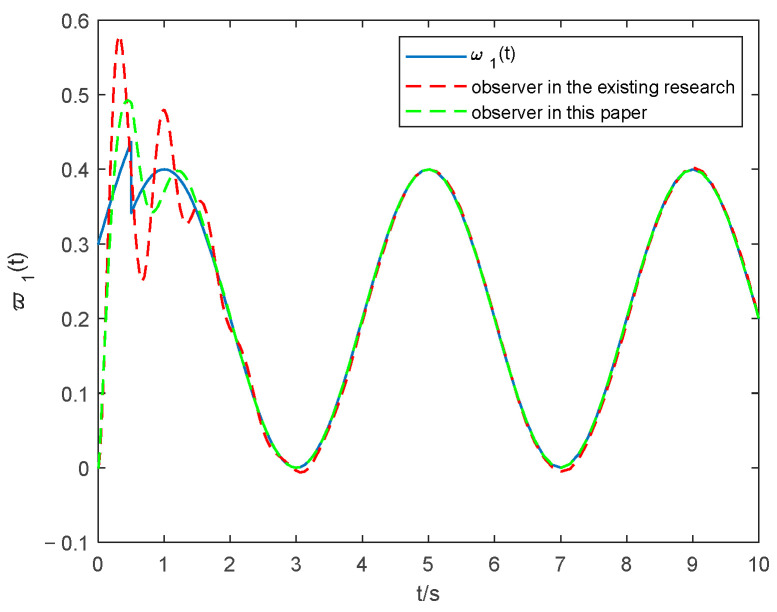
Estimation of the lumped faults $\omega_{1}$ with the observer in [30] and that in this paper.

**Figure 3 entropy-24-01068-f003:**
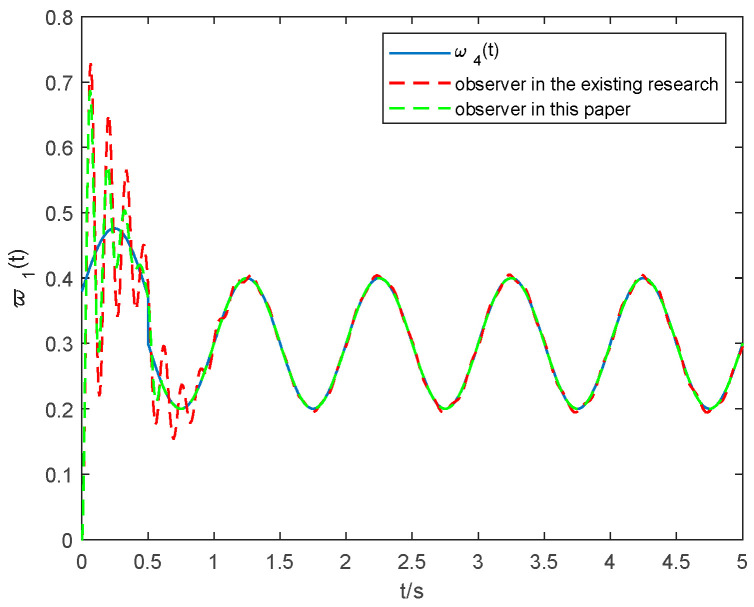
Estimation of the lumped faults $\omega_{4}$ with the observer in [30] and that in this paper.

**Figure 4 entropy-24-01068-f004:**
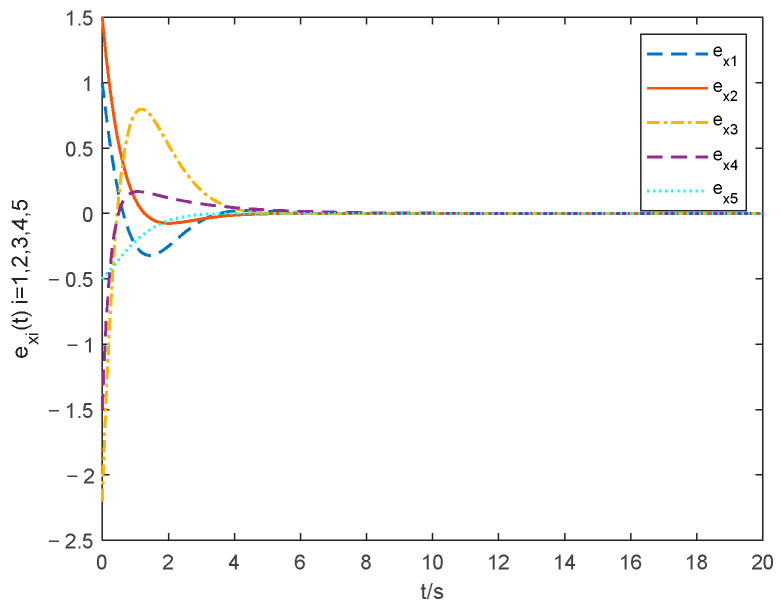
Position tracking errors with the method in this paper.

**Figure 5 entropy-24-01068-f005:**
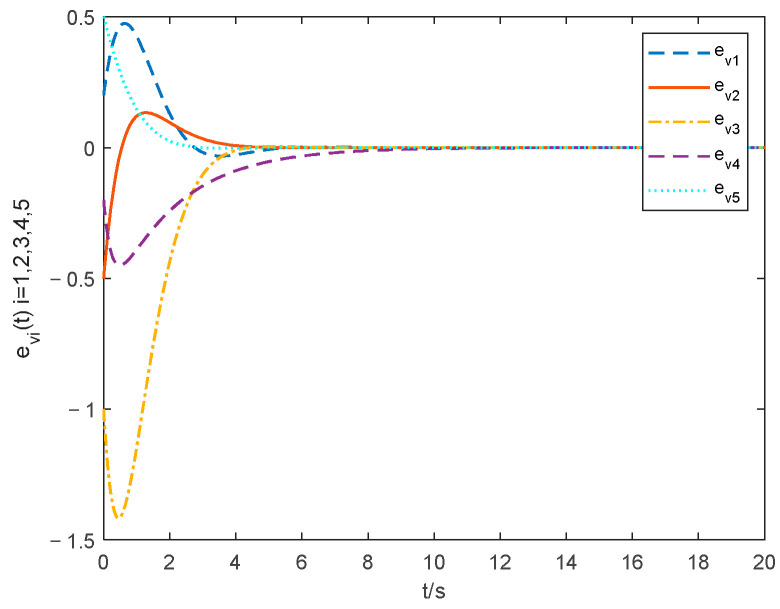
Velocity tracking errors with the method in this paper.

**Figure 6 entropy-24-01068-f006:**
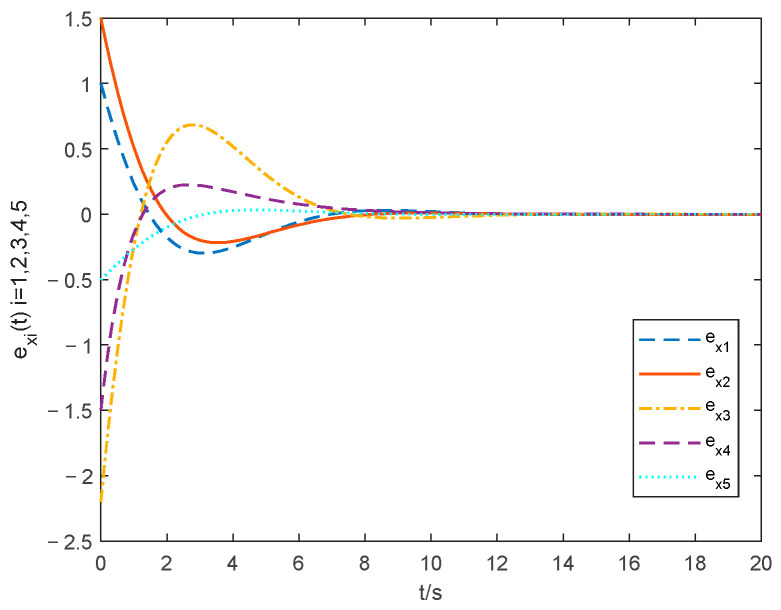
Position tracking errors with the method in [40].

**Figure 7 entropy-24-01068-f007:**
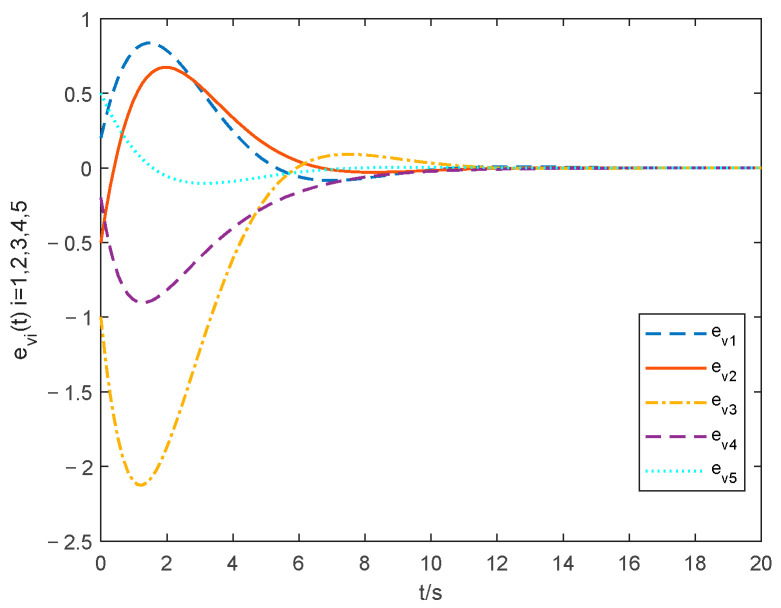
Velocity tracking errors with the method in [40].

**Figure 8 entropy-24-01068-f008:**
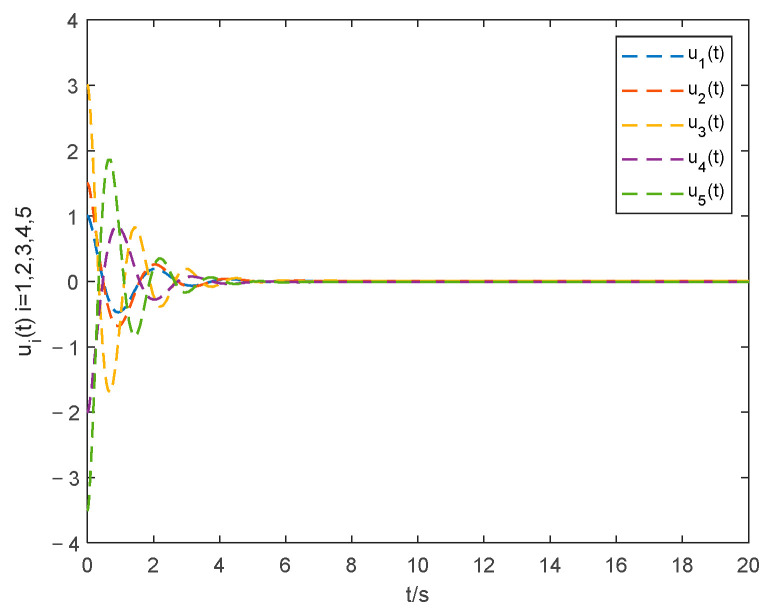
Control input with the method in this paper.

**Figure 9 entropy-24-01068-f009:**
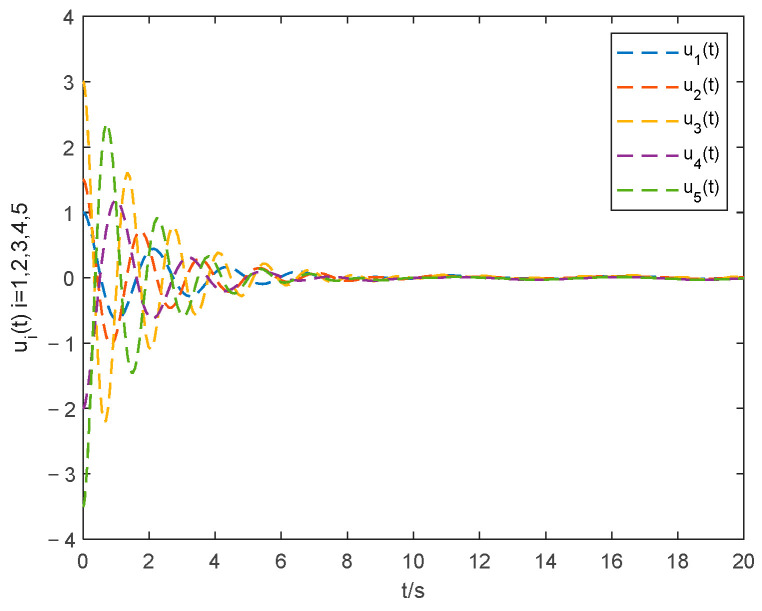
Control input with the method in [40].

**Table 1 entropy-24-01068-t001:** Estimated time of the two methods.

Method	$\mathrm{Estimated}\;\mathrm{Time}\;\mathrm{for}\;\omega_{1}$	$\mathrm{Estimated}\;\mathrm{Time}\;\mathrm{for}\;\omega_{4}$
Proposed in this paper	1.864 s	0.583 s
Proposed in [30]	2.543 s	1.064 s

## Data Availability

Not applicable.

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
