# Peer review of "Integral Non-Singular Terminal Sliding Mode Consensus Control for Multi-Agent Systems with Disturbance and Actuator Faults Based on Finite-Time Observer"

_entropy, 2022, doi:10.3390/e24081068_

Round 1

Reviewer 2 Report

It is a good paper and it can be considered for publication after some

important revisions.

Integral Non-singular Terminal Sliding Mode Consensus Control for Multi-agent Systems method is suggested in this paper for Disturbance and Actuator Faults based on Finite-time Observer.

- Some new techniques in sliding mode consensus control for multi agent systems can be considered and added to the literature study

- The suggested future work in this research field can be introduced in the last part of the conclusion.

- Please check again the paper for typos.

- The importance of the problem considered in this paper should be further addressed.

- No issue regarding the complexity of the proposed method has been presented.
